# Anodal M1 tDCS enhances online learning of rhythmic timing videogame skill

**Anthony W. Meek[1]** *, **Davin R. Greenwell** [1], **Hayami Nishio** [2], **Brach Poston[3]**,
**Zachary A. Riley[1]**

**1** School of Health and Human Sciences, Indiana University Indianapolis, Indianapolis, IN, United States of America, **2** Department of Human Physiology, University of Oregon, Eugene, WA, United States of America, **3** Department of Kinesiology and Nutrition Sciences, University of Nevada Las Vegas, Las Vegas, NV, United States of America

\* ant.meek31@gmail.com

**Data Availability Statement:** All of our data files are available from the BioStudies database (accession number S-BSST1355.)

## Abstract

Transcranial direct current stimulation (tDCS) has been shown to modify excitability of the primary motor cortex (M1) and influence online motor learning. However, research on the effects of tDCS on motor learning has focused predominantly on simplified motor tasks. The purpose of the present study was to investigate whether anodal stimulation of M1 over a single session of practice influences online learning of a relatively complex rhythmic timing video game. Fifty-eight healthy young adults were randomized to either a-tDCS or SHAM conditions and performed 2 familiarization blocks, a 20-minute 5 block practice period while receiving their assigned stimulation, and a post-test block with their non-dominant hand. To assess performance, a performance index was calculated that incorporated timing accuracy elements and incorrect key inputs. The results showed that M1 a-tDCS enhanced the learning of the video game based skill more than SHAM stimulation during practice, as well as overall learning at the post-test. These results provide evidence that M1 a-tDCS can enhance acquisition of skills where quality or success of performance depends on optimized timing between component motions of the skill, which could have implications for the application of tDCS in many real-world contexts.

## Introduction

Motor learning is fundamental in everyday life for acquiring and honing skills that range in complexity from relatively simple, like reaching and pressing a button, to more complex skills requiring coordinated sequential actions such as learning a piece of music on an instrument or throwing a ball. Motor skill improvements can accrue during a single practice session (online) or after practice is completed (offline), with both contributing to long-term retention over time [1]. The online and offline skill changes comprise the fast and slow stages of learning, with fast stage learning occurring early on, markedly during skill acquisition, and slow stage occurring later with incremental gains over multiple practice sessions [2,3]. The motor learning process is underpinned by neuroplastic changes across a spatially distributed network of interconnected brain regions [4]. Which areas are involved and the extent to which plasticity

**Funding:** The author(s) received no specific funding for this work.

**Competing interests:** The authors have declared that no competing interests exist.

within them may subserve learning, largely depends upon the characteristics of task and the stage learning [5].

The primary motor cortex (M1) appears to be crucial in the early stages of learning for fast and precise motions associated with skillful performance [6,7]. When learning a new motor skill, task-associated neurons are recruited and their repeated, sequential activation drives plastic changes synaptic strength, reinforcing connectivity [8,9]. This change in corticomotor excitability has been previously demonstrated using transcranial magnetic stimulation (TMS) in conjunction with electromyography (EMG) [10,11]. Even short bouts of skill practice (10 to 30 minutes) have been shown to elicit acute increases in M1 excitability [11,12]. Although these early, practice-related changes are relatively transient in the short term [13], it is believed that this is an important initial step of fast motor learning and skill acquisition [2,11]. Based on these short-term mechanisms as well as the more permanent structural and functional cortical reorganization associated with extensive practice and skill expertise [10,14–16], M1 is a site of particular interest for investigating motor learning.

Transcranial direct current stimulation (tDCS) is a non-invasive brain stimulation technique which is proposed to modulate cortical excitability by passing a weak, sub-threshold electrical current between two electrodes placed on or near the scalp [17]. This current passes between electrodes by passing through the surface tissues and into at underlying cortical surface [18]. The outcome of this stimulation depends on the placement and directional flow of current between electrodes with anodal tDCS (a-tDCS) typically being regarded as facilitatory, while cathodal tDCS (c-tDCS) is generally regarded as inhibitory [19]. Studies have shown that a-tDCS is able to enhance cortical excitability and LTP-like plasticity in M1 and improve motor function [20–22]. Additionally, tDCS has been shown to be particularly relevant in the acquisition and early consolidation phase of motor learning and evidence suggests that it is safe for facilitating motor learning in healthy individuals [22,23] as well as those suffering from neurological disorders [24,25].

In a recent meta-analysis by Patel and colleagues (2018), the application of tDCS to M1 during skill practice was found to enhance performance of several upper limb motor tasks through changes in speed (reaction or movement time), accuracy, the relationship between those two qualities, or reduced variability [21]. However, much of the tDCS literature reporting positive effects of M1 stimulation on performance have used simple laboratory-based tasks, such as the serial reaction time task (SRTT), sequential finger tapping task (SFTT), and sequential visual isometric pinch force task (SVIPT) that, while highly controlled, fail to represent the complexity of real-world movements [21,22,26]. Despite the non-generalizability of such findings to more complex movements, these studies lend support to M1's important role in the fast stage of learning and as a potential as a target for tDCS.

Recently, research has shown that M1 tDCS may be able to enhance performance and learning in more complex tasks like surgical skills and or videogames [27,28]. Additionally, our lab recently demonstrated that a-tDCS of the non-dominant M1 enhances online learning in both a dexterous, fine-motor tweezer task [29] as well as a complex multi-joint dart throwing task [30] performed with the non-dominant arm. Though these sorts of tasks are more generalizable to real-world skills, performance outcomes are more variable and harder to objectively measure than simpler laboratory tasks. Videogames, which share traits of both conventional laboratory motor tasks and more dynamic real-world movements, are a novel way to study the effects of tDCS on motor learning. Videogames involve diverse combinations of perceptual, attentional, cognitive, and motor skills, and practice can lead to training-induced learning [31,32]. Although there are reports of competitive gamers using tDCS to enhance performance [33], the effect of M1 tDCS in the context of gaming-based motor skill has only recently been initially investigated [27].

Our study seeks to further investigate the effects of M1 tDCS on the early stages of learning a complex videogame task. Given the task dependency of tDCS effects for relatively simple tasks [21,34,35], it is unclear how M1 tDCS will influence the acquisition of a videogame task with rhythmic sequence tapping movements combined with complex visuomotor and auditory processing demands. Based upon our previous work showing that a-tDCS of M1 enhances the early stages of learning in complex motor tasks [29,30] performed with the non-dominant arm, we hypothesized that a-tDCS of M1 would also enhance learning in a complex videogame task performed with the non-dominant arm. Our investigation was largely exploratory as, to our knowledge, this was the first study to utilize a rhythmic, sequence tapping videogame to study the effects of tDCS on motor learning.

## Methods

### Participants

A total of 60 volunteers were recruited for this study ($n$ = 30 per group; age 22.27 ± 2.78 yrs). Of these, a total of 58 were included in analysis. One subject's data excluded due significant outliers in performance scores and another was excluded due to a software issue during data collection which resulted in incomplete data. All participants were free of neurological or musculoskeletal impairments that could impact performance of the task and any volunteers who were taking medications that could influence learning, such as stimulants for ADHD, were excluded from participating in the study. Handedness was determined with the Edinburgh Handedness Inventory (EHI) [36]. Right-hand dominance was identified with scores > +40, left-hand dominance with scores < -40, while scores ≥-40 and ≤+40 were ambidextrous. In cases of ambidextrousness, the subject's preferred writing hand was considered their dominant. *(See Table 1 below for subject characterization).*

This was a prospective study which utilized a randomized, single-blinded, between-subjects, SHAM controlled design. Using MathWorks MATLAB software, a randomly generated number sequence was created and used during screening to assign subjects into either the anodal tDCS group (a-tDCS) or SHAM stimulation group (SHAM) based upon their order of recruitment. Subjects were blinded to the condition that they received for the duration of testing. The two groups were relatively matched for age (a-tDCS = 22.70 ± 2.96 yrs and SHAM = 22.34 ± 2.64 yrs) and sex (a-tDCS = 14 males, 15 females; SHAM = 15 males, 14 females). For handedness, considering preferred writing hand in the case of an ambidextrous inventory result, each group had 25 right-handed individuals and 4 left-handed individuals.

**Table 1. Subject characterization.**

| Condition | Handedness | Male | Female |
|---|---|---|---|
| **a-tDCS** | | **14** | **15** |
| | Right | 12 | 14 |
| | Left | 2 | 1 |
| **SHAM** | | **15** | **14** |
| | Right | 11 | 13 |
| | Left | 4 | 1 |

a-tDCS = Anodal transcranial direct-current stimulation.

SHAM = Sham transcranial direct-current stimulation.

All procedures were approved by Indiana University's IRB and conducted according to the Declaration of Helsinki.

## Procedures

Participants completed a single testing session while receiving a-tDCS or SHAM stimulation over M1 contralateral to the non-dominant (active) hand. During the session, subjects performed a timing-based dexterous video game task with the non-dominant hand. The protocol included 2 familiarization pre-test blocks, 5 practice blocks while receiving either a-tDCS or SHAM, and a post-test block 5 min after cessation of practice. Familiarization pre-test scores were used to determine whether the task difficulty was adequate to demonstrate motor learning during the practice period. If the task was too easy or too hard based on the familiarization trial performance, the difficulty was increased or decreased accordingly in the video game settings, and one additional familiarization trial was given at the new speed (Fig 1B).

## Step Mania task

Step Mania (https://www.stepmania.com/) is an open-source videogame where sequences of directional arrow icons (up, down, left, right) scroll upwards toward stationary arrow silhouettes (Fig 1A) and the objective is to press the appropriate arrow keys whenever a corresponding scrolling icon is perfectly aligned with (i.e. centered on) its silhouette.

Participants sat at a desk with a keyboard positioned with the arrow keys in front of the non-dominant hand. Subjects were instructed to only use one hand to tap the keys and to only tap keys one time per cue. However, no explicit instructions were given assigning specific digits to particular keys. A brief demonstration of the gameplay was shown to the participants as the objective was explained (*i.e.* press correct keys with optimal timing and avoid unnecessary/incorrect key strokes).

The same 219 cue pattern was used for all test blocks. One of the 219 cues required the left and right arrow key to be struck simultaneously, thus the pattern consisted of 55 up, 56 down, 55 left and 54 right key strokes (220 total key taps). This could not be changed through the available settings. Each testing block comprised one complete pattern. Each keystroke was categorized relative to a time window centered (time = 0s) on perfect overlap of the scrolling icon and its stationary silhouette. These timing windows were built into the program as flawless (0 to ± 0.0225s), perfect (±0.0225 to ±0.045s), great (±0.045 to ±0.090s), good (±0.090 to

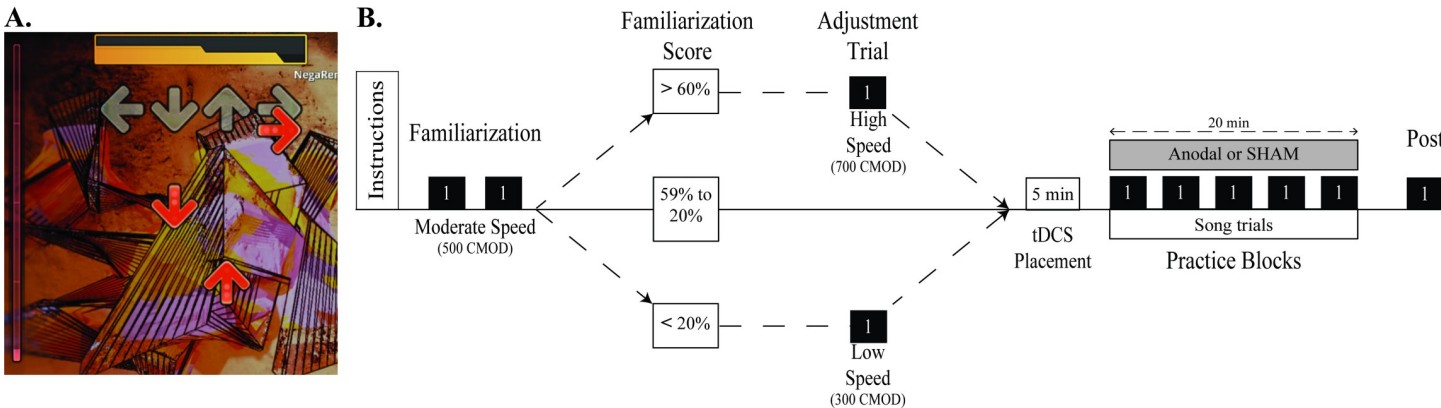

**Fig 1. Step mania overview & study timeline.** (a) Screen shot of Step Mania gameplay showing scrolling arrow icons and stationary arrow silhouettes. (b) Study procedures timeline.

±0.135s), boo (±0,135s to ± 0.180s), and miss (> ± 0.180s). To note, the first two time windows (flawless and perfect) were 22.5ms, while the other windows were 45ms.

**Baseline assessment and testing.** Subjects performed 2 familiarization trials with 500 continuous modifier (CMOD) difficulty (sets the arrow scroll speed; "arrow heights" moved per minute). An initial proficiency score was calculated (Eq 1) indicative of the number of cues being met with a tap within scoring distance. This allowed for difficulty adjustment to ensure that the task was appropriately difficult for performance gains to be demonstrated within a single practice session. It was determined in pilot testing that if too many cues were entirely missed or nearly unscored based on the inbuilt timing windows, the task overwhelmed participants and they struggled to improve. Conversely, individuals who, in their familiarization trials demonstrated very precise timing and few misses, had little room for growth.

### Familiarization Proficiency

$$FP = \left(\frac{(Flawless*1) + (Perfect*0.75) + (Great*0.5) + (Good*0.25) + (Boo*0) + (Miss*-1)}{219}\right)*100 \quad (\text{Eq 1})$$

For familiarization proficiency scores between 20 and 60, the default 500 CMOD was maintained throughout testing. For scores > 60 or < 20, the scrolling speed was adjusted up or down by 200, respectively (Fig 1B). The music tempo (210 bpm) was not affected by the CMOD change, it only affected the cue scrolling speed. Skill-adjusted subjects performed an additional familiarization block at the new speed that was used for all subsequent blocks. The last familiarization block was used as the pre-test measure for all subjects.

After familiarization and pre-test, tDCS electrodes were positioned during a 5 min break. Subjects then completed a 20 min practice period while they received either a-tDCS or SHAM during 5 practice blocks with 2 minutes of rest between blocks. The stimulation electrodes were then removed and subjects completed a post-test trial 5 minutes after completing practice.

**Evaluating performance.** At the end of each trial, the game provides feedback in the form of a numerical "game score" as well as an aggregate overview of the categorization of each keystroke timing (i.e. miss, boo, good, great, perfect, and flawless). However, this fails to account for any excess or aberrant keystrokes that fall outside of the inbuilt timing windows. For this reason, we opted to collect additional data using a separate keystroke logging software. This allowed us to devise an alternate scoring mechanism, or performance index (PI), that better reflects changes in task performance.

Two aspects of Step Mania performance are related to skill–hitting only the correct inputs and hitting them at the correct time. Both are interrelated as qualities of the task, but a change in one does not necessarily precipitate a proportional change in the other. With the guiding premise that newly acquired movement sequences are segmented, inaccurate, and jerky, whereas learned sequences are cohesive, accurate and smooth [6], a PI was calculated to incorporate incorrect inputs (Key Error Rate, KER) and temporal accuracy (2 values: Temporal Accuracy, TA; Tap Distribution Ratio, TDR) into a single value to reflect the skill associated with a certain quality of performance:

### Performance Index (PI)

$$PI = (TA - KER) \times TDR \quad (\text{Eq 2})$$

A description is provided below for each of the constituent variables in the PI equation and

how they are derived from the data provided by the game. Example data are shown after each equation to illustrate each step of the process.

**Temporal Accuracy (TA).** TA was quantified by assigning point values to the timing windows, so that "flawless" keystrokes awarded 1.0 pt and each subsequent window awarded 0.2 pts less, resulting in 0 pts for each "miss". TA provided a base value reflecting temporal accuracy of the entire 220 constituent inputs of each trial.

This is different from the Familiarization Proficiency equation we used to evaluate initial ability because we did not want to penalize a "miss" with a negative score and instead just not award any points in that case. This distinction was made because in the initial familiarization, if the speed of the game was too fast for the subject they would get overwhelmed and miss several arrows in a row without even attempting any inputs, in which case a measure was needed to clearly inform us to slow the game down. In the case of the timing accuracy measure (*Eq 2*), the subject would typically only miss an arrow occasionally and therefore just not be awarded points for it.

## Temporal Accuracy (TA)

$$TA = ((Flawless*1) + (Perfect*0.8) + (Great*0.6) + (Good*0.4) + (Boo*0.2) + (Miss*0)) \tag{Eq 3}$$

For a subject with 72 Flawless, 69 Perfect, 58 Great, 10 good, 3 Boo and 7 miss:

$$TA = (72*1) + (69*0.8) + (58*0.6) + (10*0.4) + (3*0.2) + (7*0) = 166.6$$

**Key Error Rate (KER).** KER evaluated execution errors such as pressing multiple keys simultaneously, tapping the same key several times, or pressing a cycle of keys 'searching' for the correct one. Imprecise and unstable finger movements relate to learning in aspects of the skill, irrespective of timing (i.e. key sequence order, hand positioning, etc). Utilizing the extra keystroke data collected during the game, the differences between actual key stroke totals and the number of times the arrow keys appear in the sequence were calculated for each trial and used in Eq 3 to produce a scaled error value relative to the total number of cues.

## Key Error Rate

$$KER = \left(\frac{(up - 55) + (down - 56) + (left - 55) + (right - 54)}{220}\right)*100 \tag{Eq 4}$$

## For a subject with 56 up, 57 down, 53 left and 53 right arrow:

$$KER = \left(\frac{(|56 - 55|) + (|57 - 56|) + (|53 - 55|) + (|53 - 54|)}{220}\right)*100 = 2.273$$

**Tap Distribution Ratio (TDR).** In rhythm gaming, ratios of the tap totals in the different timing windows are commonly used in conjunction with timing scores because, as a consequence of binning taps into scored timing windows that are relatively wide, attempts can yield arbitrarily similar TAs despite the distribution of taps in the windows reflecting different levels of play. We adapted this strategy to further differentiate skill in the task and enhance sensitivity

in the measure to changes in the quality of performance. TDR adjusts scores based on the concentration of taps in the two best and two worst categories, with additional weighting based on the proportion of those that are in the best (Flawless; most skillful) and worst (Miss; least skillful) timing windows. To properly illustrate the function of this variable, the original TA data from above will be used as well as a second example that produces an identical 166.6 TA value.

### Tap Distribution Ratio (TDR)

$$TDR = \frac{\left(1 + \left(\frac{Flawless + Perfect)}{219}\right)\right) * \left(1.5 + \left(\frac{Flawless}{219}\right)\right)}{\left(1 + \left(\frac{Boo + Miss}{219}\right)\right) * \left(1.5 + \left(\frac{Miss}{219}\right)\right)} \qquad \text{(Eq 5)}$$

For a subject with 72 Flawless, 69 Perfect, 58 Great, 10 good, 3 Boo and 7 Miss:

$$TDR = \frac{\left(1 + \left(\frac{(72+69)}{219}\right)\right) * \left(1.5 + \left(\frac{72}{219}\right)\right)}{\left(1 + \left(\frac{3+7}{219}\right)\right) * \left(1.5 + \left(\frac{7}{219}\right)\right)} = 1.877$$

For a second subject with 82 Flawless, 46 Perfect, 67 Great, 18 Good, 2 Boo, 4 Miss:

$$TDR = \frac{\left(1 + \left(\frac{(82+46)}{219}\right)\right) * \left(1.5 + \left(\frac{82}{219}\right)\right)}{\left(1 + \left(\frac{2+4}{219}\right)\right) * \left(1.5 + \left(\frac{4}{219}\right)\right)} = 1.904$$

Assuming a similar KER value for both of these trials, the PI for each would be:

$$PI = (TA - KER) * TDR$$

$$A.) \; PI = (166.6 - 2.273) * 1.877 = 307.32$$

or

$$B.) \; PI = (166.6 - 2.273) * 1.904 = 311.7$$

In utilizing a TDR factor to modify PI scores, we are able to better reflect the more subtle changes in subject performance that might otherwise be missed when relying on TA and KER values alone.

### tDCS

A Soterix Medical 1x1 Low Intensity transcranial DC stimulator was used to deliver a-tDCS with parameters previously determined to be effective and safe (duration 20 min; current 1mA; active electrode over M1 contralateral to the non-dominant hand and reference electrode over the ipsilateral supraorbital). These parameters were chosen because previous research, including work from our lab, has shown that 1mA of current delivered for 20 minutes is sufficient to elicit changes in cortical excitability and motor performance and that higher doses (longer durations and/or higher intensities) may result in diminishing or inverse effects [37–41]. Current was delivered through a pair of 5cm x 5cm (25cm$^2$) rubber electrodes which were placed inside of saline-soaked sponges and affixed to the head with rubber straps. In the a-tDCS condition, the current intensity was ramped up over the course of 30s, maintained at 1mA for 20 minutes, and followed by a 30s ramp down. For the SHAM condition, a 30 second ramp up and then down was performed at the beginning and end of the 20-minute period to simulate the sensation of stimulation.

## Data analysis and statistics

While we recognize the importance of performing an a priori sample size calculation, our study was largely exploratory, and no effect size of stimulation or task performance score could be estimated from previous literature. For this reason, no initial sample size calculation was performed. Instead, we estimated a sample size based on our two previous studies which showed significant effects of a-tDCS on motor task performance in a non-dominant hand tweezer pegboard task (n = 40) [37] and a non-dominant hand, dart throwing task (n = 58) [38]. Notably, between the dart throwing and tweezer task, the dart throwing was characterized by more degrees of freedom and, in turn, more task performance variability. Since the Step Mania task involved less upper arm movement and fewer degrees of freedom than our dart throwing task but more cognitive and technical complexity than our tweezer task, we determined that an appropriate recruitment target would likely lay somewhere between the 40 and 58 subjects. Still, because we were unsure, we opted to recruit 60 with the hopes that this number would provide us with sufficient statistical power for our analyses.

A PI value was calculated for each trial. PI gain scores (block PI–baseline PI) were calculated for each practice block as well as the post-test and used observe the change in performance over time relative to baseline performance. Normality was tested with a Shapiro-Wilk test. Baseline PI scores were compared between groups with an independent *t*-test to test for baseline differences before gain scores were calculated.

Separate comparisons were performed for the practice blocks and the post-test block. A mixed ANOVA (2 groups x 5 blocks) with repeated measures on block was used to compare the practice gain scores and effect size were calculated as partial eta squared ($\eta_p^2$). For the post-test block, an independent *t*-test was used to compare the gain scores between the conditions and effect size was calculated as Cohen's *d*. A p-value of 0.05 was considered statistically significant. Bonferroni post-hoc tests were used to determine differences for multiple comparisons. Data are presented as mean ± standard deviation in text and as mean ± standard error in figures. All data analysis was performed with SPSS 24.

## Results

### Practice blocks

Over the course of practice (B1 through B5), both groups gradually improved TA (range: 147.78 to 160.72) and TDR also modestly increased (range: 1.56 to 1.80). The rate of errors (KER) also improved (i.e. reduced) across practice for both groups (range: 3.83 to 2.71) (see Table 2). The calculated PI score for each block incorporated all 3 of these metrics.

Practice block PI gain scores were calculated relative to baseline to evaluate how overall performance of the two groups changed during practice. An independent *t*-test on the baseline PI values showed a-tDCS (181.80 pts ± 66.04) and SHAM (188.97 pts ± 79.39) performance was similar prior to practice (*t(56) = -0.37, P = 0.71*). An a-tDCS subject was missing error data for two practice trials and thus was excluded from the practice block analysis (a-tDCS *n* = 28, SHAM *n* = 29).

A Shapiro-Wilk test showed normally distributed gain scores (*P = 0.278 to 0.977*) for all cells of the design except a-tDCS B1 (*P = 0.004*) and a-tDCS B4 (*P = 0.01*). Since ANOVA type 1 error rate is fairly robust to normality violations [42–44] the mixed ANOVA was still performed. Homogeneity of variances was confirmed by Levene's test (*P = 0.25 to 0.65*). A Huynh-Feldt correction was used to adjust for a sphericity violation (*P = 0.003*). There was a statistically significant interaction between the stimulation group and block on practice gain scores (*F[3.466, 190.605] = 3.042, P = 0.024, $\eta_p^2$ = 0.052*). Post hoc analysis of the simple main

**Table 2. TA, KER, & TDR scores across time.**

| | | Practice | | | | | |
|---|---|---|---|---|---|---|---|
| | **Baseline** | **B1** | **B2** | **B3** | **B4** | **B5** | **Post-Test** |
| **a-tDCS** | | | | | | | |
| TA | 135.66(19.81) | 149.84(19.16) | 152.85(20.02) | 154.31(15.99) | 159.42(15.04) | 160.72(16.01) | 166.63(15.57) |
| KER | 5.36(4.53) | 3.83(3.42) | 3.30(2.59) | 3.15(2.06) | 3.26(2.33) | 2.82(1.91) | 2.63(1.93) |
| TDR | 1.35(0.29) | 1.59(0.32) | 1.65(0.34) | 1.66 (0.28) | 1.76(0.28) | 1.80(0.31) | 1.92(0.31) |
| **SHAM** | | | | | | | |
| TA | 134.96(27.41) | 147.78(22.42) | 150.78(22.32) | 154.21(19.79) | 152.66(17.71) | 155.36(19.73) | 159.62(18.98) |
| KER | 3.82(2.99) | 3.32(1.97) | 2.74(2.33) | 2.90(2.16) | 2.98(2.30) | 2.71(1.82) | 2.03(1.79) |
| TDR | 1.37(0.36) | 1.56(0.35) | 1.62(0.35) | 1.68(0.35) | 1.64 (0.32) | 1.69(0.37) | 1.78(0.37) |

TA - Temporal Accuracy; KER - Key error rate; TDR - Tap Distribution Ratio.

effects for group revealed a significant difference between conditions at B4 ($P$ = 0.008) and B5 (*P = 0.046*), where a-tDCS (B4: 100.02pts ± 47.21; B5: 109.05pts ± 60.92) resulted in greater PI improvement than SHAM (B4: 62.05pts ± 55.75; B5: 76.58pts ± 59.16). Post hoc analysis for block showed that B3 and B5 SHAM gain scores were significantly different from B1 (*P = 0.002* and *P = 0.005*). For a-tDCS, B4 and B5 gain scores were statistically significantly higher than blocks 1 (both, *P < 0.001*), 2 (*P = 0.049* and *P = 0.008*) and 3 (*P = 0.005* and *P = 0.002*).

### Pre-test to post-test

One SHAM subject was missing error data for the post-test and thus was excluded from analysis (a-tDCS $n$ = 29, SHAM $n$ = 28). Normality of the gain scores was confirmed with a Shapiro-Wilk test (*P = 0.661* to *0.540*). An independent $t$-test on the gain scores from baseline to post-test showed a statistically significant difference between conditions (*t(55) = 2.234*, *P = 0.030*, *Cohen's d = 0.592*), with a-tDCS (137.844pts ± 66.218) leading to greater improvement compared to SHAM (99.501pts ± 63.28) (Fig 2).

### Discussion

Our results indicate that, compared to SHAM stimulation, M1 a-tDCS has a beneficial effect on the acquisition of a dexterous timing-based video game. Task improvement was indicated by higher gain scores, representing increased cumulative performance with better temporal accuracy and reduced error rate. Although both conditions demonstrated significant differences in performance across practice, the level of improvement with a-tDCS at B4 and B5 was significantly greater than that attained by SHAM (Fig 2). Likewise, considering only the change in performance from baseline to post-test, a-tDCS during practice resulted in a greater overall change in PI compared to SHAM stimulation (Fig 2).

While the a-tDCS group showed greater improvements than the SHAM group in their global PI score, this metric, alone, lacks detail about how execution of the SM task was changed to improve performance. Visual inspection of the general raw data trends (TA windows and KER) suggests that changes in both speed and accuracy elements of the task were factors driving and differentiating performance gains between the groups (Fig 3A–3C). Despite what appear to be disparities in baseline KER scores between groups (Fig 3C), these differences were non-significant (*p = 0.149*), resulted in only marginal (∼7 points) differences in the baseline PI scores between groups, and were largely attenuated by the end of the initial practice block. Since both groups demonstrated rapid initial KER changes with similar trends across practice

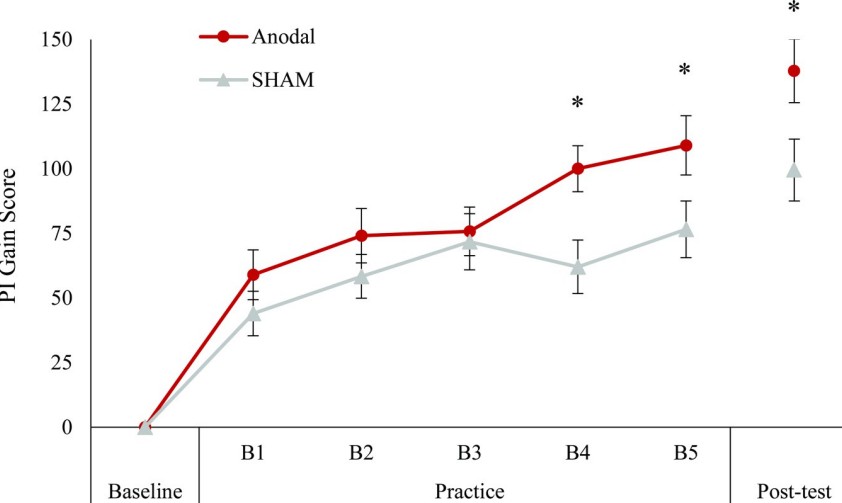

**Fig 2. Change in Performance Index (PI) gain score across time.** Data show the practice block PI gain scores relative to baseline (i.e. last familiarization block) for a-tDCS (*n* = 28) and SHAM (*n* = 29) conditions. The gain scores from baseline to the post-test block for a-tDCS (*n* = 29) and SHAM (*n* = 28) are also shown. Whiskers denote the standard error. Both groups displayed significant increases in performance across practice. a-tDCS gain scores were significantly greater than SHAM at B4 and B5. At the post-test, a-tDCS resulted in significantly greater gains in performance compared to SHAM. *Indicates significant difference from SHAM ($P < 0.05$).

and post-test (Fig 3C), it can be inferred that both groups acquired the sequence order similarly. Additionally, both a-tDCS and SHAM groups had more taps in the best TA window with a concurrent decrease in the worst window (Fig 3B). However, by practice B5 and the post-test, there is a clear TA advantage favoring a-tDCS with 10.9 and 13.5 more taps on average than SHAM in the best two timing windows combined and 3.1 and 2.3 fewer misses in B5 and the post-test, respectively (Fig 3A). These results suggest that the greater performance improvements within the a-tDCS group may have been attributed to the enhanced "chunking" of Step Mania subsequences,

The chunk formation process can be thought of as a transition from high uncertainty to low uncertainty in the execution of a motor skill that leads to sequence accuracy increases as the transition occurs [45]. In early stages of sequence learning with internally cued movements, efficient learning appears to prioritize learning of spatial elements and then shift to temporal components [45]. Previous work has demonstrated that M1 tDCS can accelerate chunk formation in early motor learning within the first training session [46]. Our results support this pattern for externally cued sequences as well, as evident by the quick initial improvements in KER and the similar learning curve in both groups (Fig 3C). Whether timing-based performance differences resulted from enhanced encoding of temporal sequence elements across the motor network or enhanced local M1 circuitry (enlarged representation or improved activity pattern), the behavioral output that increased PI scores resulted from M1 a-tDCS, which supports the important role of M1 for fast motor learning and as a sequence learning substrate [47].

It is also possible that tDCS may have augmented Step Mania skill acquisition by improving the transitions between taps through the enhancement of movement coarticulations. Coarticulation is the tendency for one element of a movement sequence to be generated in a manner which facilitates preceding or subsequent elements [48]. In our study, this was observed in subjects who appeared to develop anticipatory finger motions which allowed them to prepare for subsequent key presses. It is therefore plausible that, through enhanced chunking of

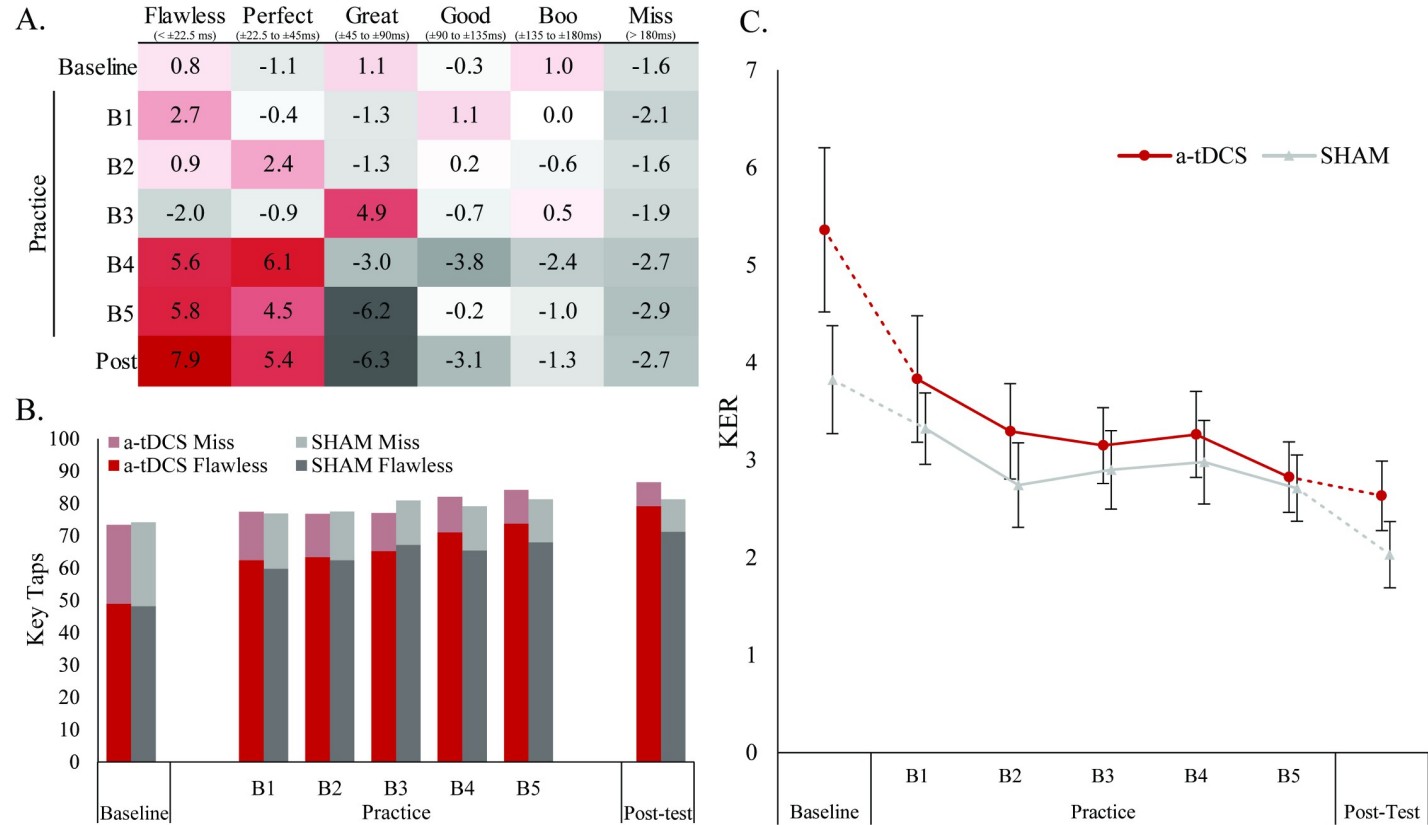

**Fig 3. Change in subject performance characteristics.** Data illustrate changes in the concentration of taps in the timing windows across all trials (A, B) and the key error rates (C) for both conditions. The color gradient table (A) shows the difference between the average a-tDCS tap distribution and average SHAM tap distribution (a-tDCS total–SHAM total) for each trial. Darker red tones signify differences favoring a-tDCS (positive values) and darker grey tones signify differences favoring SHAM (more negative). The stacked bar graph (B) shows the total number of taps in the best ("Flawless", 0 to ± 0.0225s) and worst ("Miss", > ± 0.180s) timing windows across trials. The total height of the bars represents the combined total number of taps falling within those two windows. The dark bottom portions of the bars indicate the proportion of flawless taps and the lighter top portions represent misses. The line graph (C) shows the KER trend from baseline, across practice to the post-test. Whiskers denote standard error.

movement sequences, M1 tDCS accelerated the emergence of specific coarticulations which facilitated more rapid and accurate transitions between taps.

Compared with SHAM, improved task component chunking and enhanced movement coarticulation in the M1 tDCS group appears to have resulted in superior relative timing of SM task performance. This is important because relative timing, appears to be critically important in motor skill learning and performance in rhythm based tasks [49,50]. Apolinário-Souza and colleagues (2016) previously demonstrated that M1 tDCS only enhances absolute timing [51], however, taken in the context of our current findings, we believe that M1 tDCS effects on relative timing may be dependent on task complexity. Apolinário-Souza and colleagues observed contextual changes during a single-finger transfer task, while our study analyzed the effects on learning under the practiced conditions with a comparatively complex multi-finger task. This is a particularly relevant distinction because it has been previously shown that complex movements may be more susceptible to tDCS accelerated learning than simpler tasks [52]. Overall, our results reinforce the idea that M1 is an important neural substrate for motor sequence learning, especially when the quality or outcome success of the skill relies directly upon optimized relative timing (rhythm) between the component movements (playing a

musical instrument or shooting a jump shot), rather than the fastest possible performance of correctly ordered elements (typing on a keyboard or a serial reaction timing task).

Our results align with previous findings that indicate M1 a-tDCS is a beneficial tool to enhance acquisition and performance of dexterous motor skills [21,34,53]. However, most of the previous evidence is based on simple laboratory tasks that do well isolating and measuring specific learning processes, but lack the variety of parameters many real-world motor skills involve. Step Mania is a complex rhythmic timing video game that provides ample explicit feedback. Even though M1 is important in skill acquisition, the effects of tDCS on motor learning are known to be task-dependent [34,35] and when several areas outside M1 influence task performance, it has been suggested that M1 modulation alone may be insufficient to enhance motor learning [21]. Within this context, our finding that M1 was a viable target for tDCS to enhance acquisition and performance of a novel complex task is particularly interesting.

To our knowledge this is only the second study demonstrating M1 tDCS can enhance acquisition of video gaming motor skill [27], and the first specifically within the rhythm gaming genre. Video gaming inputs (i.e. controllers, keyboard and mouse) require fine, dexterous hand movements and gross arm motor skills that, when combined with the complex visual processing and attentional demands of video games, may provide greater generalizability than traditional paradigms for studying motor learning [27]. Additionally, tDCS has reportedly begun to see commercial use (i.e. foc.us or Halo devices) as a performance enhancing tool in the lucrative market of competitive gaming and eSports [33]; however, the sensory, cognitive, and motor demands between different games can vary drastically and the ideal stimulation parameters for these different gaming demands are unclear. Though our results support and extend the evidence that M1 tDCS can improve gaming motor skills, it is not clear whether our chosen stimulation parameters are optimal or generalizable to other types of video games. Furthermore, our study only investigated the effects of a single bout of M1 tDCS on videogame performance in the non-dominant hand while most videogames involve some degree of bimanual control and regular, repeated practice. Since bimanual motor tasks are generally more complex and involve broader, network-wide activity than unimanual tasks [54–56], and since the effects of tDCS stimulation appear differ between hemispheres [57–59], it challenging to say whether or not our chosen, stimulation parameters would translate to a more complex, bimanual videogame task. Finally, while our results may also provide further indication that M1 tDCS may be able to enhance rehabilitation motor outcomes, our research was conducted only on a healthy population and further research, particularly in conjunction with video-game-based rehabilitation interventions should be conducted [60].

## Conclusion

tDCS shows promise as a motor learning adjuvant in many settings, but due to the task specific nature of tDCS effects on simplified laboratory tasks, there is little consensus on ideal stimulation targets to enhance motor learning for more complex real-world tasks. We have demonstrated that, compared to SHAM stimulation, M1 a-tDCS can significantly improve acquisition and performance of a complex dexterous video game skill within a single practice session. In contrast to simpler sequence learning tasks frequently used to study tDCS effects, Step Mania better represents real world skills where timing between component motions and processing incoming information relevant to task success are pivotal for learning and performance. In light of recent work highlighting that M1 might not be the primary site for plasticity that supports learning in traditional lab-based sequence tasks [61,62], and that lab-based tasks may not probe learning a skilled continuous sequential action with high fidelity [63], the

observed effects in the present study provide interesting evidence that supports the potential of M1 a-tDCS for practical real-world applications.

## Author Contributions

**Conceptualization:** Anthony W. Meek, Davin R. Greenwell, Hayami Nishio, Brach Poston, Zachary A. Riley.

**Data curation:** Anthony W. Meek, Davin R. Greenwell, Hayami Nishio, Zachary A. Riley.

**Formal analysis:** Anthony W. Meek, Davin R. Greenwell, Hayami Nishio, Zachary A. Riley.

**Investigation:** Anthony W. Meek, Davin R. Greenwell, Hayami Nishio, Zachary A. Riley.

**Methodology:** Anthony W. Meek, Davin R. Greenwell, Hayami Nishio, Brach Poston, Zachary A. Riley.

**Project administration:** Anthony W. Meek, Davin R. Greenwell, Brach Poston, Zachary A. Riley.

**Supervision:** Anthony W. Meek, Davin R. Greenwell, Hayami Nishio, Zachary A. Riley.

**Validation:** Anthony W. Meek, Davin R. Greenwell, Zachary A. Riley.

**Writing – original draft:** Anthony W. Meek.

**Writing – review & editing:** Anthony W. Meek, Davin R. Greenwell, Brach Poston, Zachary A. Riley.

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
