## [Decision Letter · Decision Letter 0]

14 Feb 2024

PONE-D-23-37863Anodal M1 tDCS enhances online learning of rhythmic timing videogame skillPLOS ONE

Dear Dr. Greenwell,

Thank you for submitting your manuscript to PLOS ONE. After careful consideration, we feel that it has merit but does not fully meet PLOS ONE’s publication criteria as it currently stands. Therefore, we invite you to submit a revised version of the manuscript that addresses the points raised during the review process. 

We look forward to receiving your revised manuscript.

Kind regards,

Sandra Carvalho, Ph.D.

Academic Editor

PLOS ONE

Journal Requirements:

2. For studies involving third-party data, we encourage authors to share any data specific to their analyses that they can legally distribute. PLOS recognizes, however, that authors may be using third-party data they do not have the rights to share. When third-party data cannot be publicly shared, authors must provide all information necessary for interested researchers to apply to gain access to the data. (https://journals.plos.org/plosone/s/data-availability#loc-acceptable-data-access-restrictions) 

a) A description of the data set and the third-party source

b) If applicable, verification of permission to use the data set

c) Confirmation of whether the authors received any special privileges in accessing the data that other researchers would not have

d) All necessary contact information others would need to apply to gain access to the data

3. Please amend your authorship list in your manuscript file to include author Dr. Hayami Nishio.

Reviewers' comments:

Reviewer's Responses to Questions

**Comments to the Author**

1. Is the manuscript technically sound, and do the data support the conclusions?

Reviewer #1: Yes

Reviewer #2: Yes

2. Has the statistical analysis been performed appropriately and rigorously? 

Reviewer #1: Yes

Reviewer #2: Yes

3. Have the authors made all data underlying the findings in their manuscript fully available?

Reviewer #1: Yes

Reviewer #2: No

4. Is the manuscript presented in an intelligible fashion and written in standard English?

Reviewer #1: Yes

Reviewer #2: Yes

5. Review Comments to the Author

Reviewer #1: The present study sought to evaluate the impact of M1 tDCS on a single session motor practice. The authors were very successful in choosing the research problem, study design, method and articulation of results with the literature. The work is clear, objective, detailed and as a reader, there are no doubts about the theoretical basis, method or how the data were analyzed. The work clearly makes a contribution to the literature, its reading is fluid and the reader finds it easy to reproduce its detailed method. I believe that this work should be accepted by the journal, since it will contribute significantly to the field, both in clinical practice and research in neuromodulation. Below I send 5 comments to improve the work.

Major comments

1) The authors clearly describe the literature that supports the objectives, but do not present the hypotheses in order to verify whether the results are compatible or not with the hypotheses. I suggest describing in detail and referencing the literature, the hypotheses raised a priori and relating them to the analyzes described;

2) The authors describe that they chose a sample of 58 participants, but do not describe how they arrived at this value. Furthermore, it is not clear whether there was any sample loss or even the exclusion of any participant for any reason (such as an outlier);

3) The characterization of the collected sample was not presented, for example in a table;

4) The authors do not describe the limitations of the study;

5) Although the authors have provided an excellent discussion of the results, the length of this topic is not compatible with the other topics in the work. I suggest reviewing the discussion completely, seeking to synthesize it significantly, in order to make it more objective and direct.

Reviewer #2: I have thoroughly reviewed the manuscript titled "Anodal M1 tDCS enhances online learning of rhythmic timing videogame skill" and found the study to be a valuable contribution to our understanding of motor learning, particularly within the context of complex real-world tasks. The manuscript is well-written and provides significant insights into the potential of M1 anodal transcranial direct current stimulation (a-tDCS) to enhance motor learning, especially in the domain of complex real-world tasks, such as the Step Mania video game. However, to strengthen the manuscript, I recommend considering the following points for incorporation.

Introduction:

I suggest refining the introduction to lead the reader more effectively to understand the formulation of the research question.

For instance, the authors referenced lab-based tasks on page 4, yet I missed a discussion on the other studies conducted with more complex tasks to offer additional context, since they have only mentioned an increasing number of tasks expanding complexity but not explored these studies.

It would also be valid to ensure clarity in articulating the specific study's objectives and specify hypotheses related to anticipated changes in skill learning resulting from M1 a-tDCS application.

Finally, focus the introduction more explicitly on the study’s objective. For instance, information about multiple sections (page 3, line 1-2) may be considered less pertinent than the other points I mentioned and could be streamline for better alignment with the study’s main focus.

Methods:

The method is well-written, with detailed and easily-followed methodology. I appreciate the care taken in creating a score and reporting it in such a meticulous manner. However, I would like to offer some suggestions:

Please include information on whether a sample size calculation was conducted.

Could the authors provide clarification on whether the study was retrospective? Additionally, it would be valuable to know if the data analyzed in this study has also been considered in another publication.

It would be appreciated if the authors could provide details on the randomization procedure and specify if participation was voluntary. Consider following guidelines for reporting your experiment.

Could the authors enhance the resolution of Figure 1?

An explanation of the rationale behind choosing current intensity and stimulation duration would be helpful.

Could the authors provide clarification on whether participants' naivety to rhythm games served as an control variable in the study? If not, I think it would be beneficial to include a comment on this aspect in the discussions. However, in the discussion (pag 23), it is mentioned that participants were naive to rhythm games – was this an exclusion criterion then? If so, it would be helpful to add this information to the description of exclusion criteria.

Results:

Consider incorporating the results of the post-test into Figure 2 to provide an overarching view of the experiment and the longevity of effects post-training.

Discussion:

The discussion section of the paper addresses various crucial points regarding the outcomes derived from the application of M1 a-tDCS in acquiring motor skills within a timing-based video game. Here are some suggestions to further elaborate on the discussion:

Elaborate on the possibility that the a-tDCS group started with more errors in baseline in Figure 4C.

Discuss additional limitations and suggest areas for future research.

Given that the study's sample comprises a healthy population, consider rephrasing the last sentence of the discussion (page 24, l. 14-16) to emphasize the need for additional research within a rehabilitation context.

Minor comments:

Italicize the p-values

Spell out PM and SMA before using the acronyms (pag 22)

6. PLOS authors have the option to publish the peer review history of their article (what does this mean?). If published, this will include your full peer review and any attached files.

Reviewer #1: **Yes: **Lucas Murrins Marques

Reviewer #2: No

---

## [Author Response · Author response to Decision Letter 0]

8 Mar 2024

Comments to the Author

1. Is the manuscript technically sound, and do the data support the conclusions?

Reviewer #1: Yes

Reviewer #2: Yes

2. Has the statistical analysis been performed appropriately and rigorously? 

Reviewer #1: Yes

Reviewer #2: Yes

3. Have the authors made all data underlying the findings in their manuscript fully available?

Reviewer #1: Yes

Reviewer #2: No

4. Is the manuscript presented in an intelligible fashion and written in standard English?

Reviewer #1: Yes

Reviewer #2: Yes

5. Review Comments to the Author

Reviewer #1: The present study sought to evaluate the impact of M1 tDCS on a single session motor practice. The authors were very successful in choosing the research problem, study design, method and articulation of results with the literature. The work is clear, objective, detailed and as a reader, there are no doubts about the theoretical basis, method or how the data were analyzed. The work clearly makes a contribution to the literature, its reading is fluid and the reader finds it easy to reproduce its detailed method. I believe that this work should be accepted by the journal, since it will contribute significantly to the field, both in clinical practice and research in neuromodulation. Below I send 5 comments to improve the work.

Major comments

1) The authors clearly describe the literature that supports the objectives, but do not present the hypotheses in order to verify whether the results are compatible or not with the hypotheses. I suggest describing in detail and referencing the literature, the hypotheses raised a priori and relating them to the analyzes described;

Thank you for catching this. This was an oversight on our part. We hypothesized that, compared to SHAM, a-tDCS would enhance motor task learning. This was based largely on our previous work which demonstrated that M1 targeted a-tDCS enhanced motor learning in tasks performed with the non-dominant arm (A. Meek et al., 2021; M. Wilson et al., 2022). However, at the time of conducting this study, to our knowledge, no previous literature had explored the effects of a-tDCS in a comparable videogame task. Therefore, the main purpose of our investigation was exploratory.

2) The authors describe that they chose a sample of 58 participants, but do not describe how they arrived at this value. Furthermore, it is not clear whether there was any sample loss or even the exclusion of any participant for any reason (such as an outlier);

This is a good point that we seem to have overlooked in our manuscript. While we recognize the importance of performing initial sample size calculations, our current project was largely exploratory, and we did not have any comparable literature from which to estimate a range for task performance or the effect sizes of stimulation. 

Therefore, we estimated a sample size based on our two previous studies which showed significant effects of a-tDCS on motor task performance in samples of 40 subjects in a non-dominant hand tweezer pegboard task and of 58 subjects in a non-dominant hand, dart throwing task. Notably, between the dart and tweezer task, the dart task was characterized by more degrees of freedom and, in turn, more task performance variability. Since the Step Mania task involved less upper arm movement and fewer degrees of freedom than the dart task but more cognitive and technical complexity than the tweezer task, we determined that an appropriate recruitment target would likely lay somewhere between the 40 and 58 subjects. Still, because we were unsure, we opted to recruit 60 with the hopes that this number would provide us with sufficient statistical power for our analyses. One subject demonstrated extreme outliers in performance scores and due to a hardware error, another subject’s data was partially lost. As a result, a total of 2 subjects were excluded from analysis.

3) The characterization of the collected sample was not presented, for example in a table;

Table 1: Subject Characterization

Condition Male Female

 Handedness 

a-tDCS 14 15

 Right 12 14

 Left 2 1

SHAM 15 14

 Right 11 13

 Left 4 1

a-tDCS = Anodal transcranial direct-current stimulation

SHAM = Sham transcranial direct-current stimulation

4) The authors do not describe the limitations of the study;

Thank you for pointing this out. This was an oversight on our part. We have revised the final paragraph of the discussion to include a better summary of the study’s limitations.

5) Although the authors have provided an excellent discussion of the results, the length of this topic is not compatible with the other topics in the work. I suggest reviewing the discussion completely, seeking to synthesize it significantly, in order to make it more objective and direct.

Thank you again for your insight. We have completely revised and synthesized the discussion the hopes that is more direct and to the point. We agree that there were several components that, while maybe tangentially related, were not directly relevant to the discussion at hand.

Reviewer #2: I have thoroughly reviewed the manuscript titled "Anodal M1 tDCS enhances online learning of rhythmic timing videogame skill" and found the study to be a valuable contribution to our understanding of motor learning, particularly within the context of complex real-world tasks. The manuscript is well-written and provides significant insights into the potential of M1 anodal transcranial direct current stimulation (a-tDCS) to enhance motor learning, especially in the domain of complex real-world tasks, such as the Step Mania video game. However, to strengthen the manuscript, I recommend considering the following points for incorporation.

Introduction:

I suggest refining the introduction to lead the reader more effectively to understand the formulation of the research question.

For instance, the authors referenced lab-based tasks on page 4, yet I missed a discussion on the other studies conducted with more complex tasks to offer additional context, since they have only mentioned an increasing number of tasks expanding complexity but not explored these studies.

It would also be valid to ensure clarity in articulating the specific study's objectives and specify hypotheses related to anticipated changes in skill learning resulting from M1 a-tDCS application.

Finally, focus the introduction more explicitly on the study’s objective. For instance, information about multiple sections (page 3, line 1-2) may be considered less pertinent than the other points I mentioned and could be streamline for better alignment with the study’s main focus.

Thank you for your suggestions. We have revised the introduction accordingly and believe that it now aligns better with the study’s main focus. We have attempted to remove less pertinent information and included additional details about the more complex task with which tDCS has been used to modulate motor learning. We have also added details about our specific objectives and our central hypothesis. It is our hope that these changes will provide the reader with a better context for our research and help make clear the purpose of our research.

Methods:

The method is well-written, with detailed and easily-followed methodology. I appreciate the care taken in creating a score and reporting it in such a meticulous manner. However, I would like to offer some suggestions:

Please include information on whether a sample size calculation was conducted.

While we recognize the importance of performing initial sample size calculations, our current project was largely exploratory, and we did not have any comparable literature from which to estimate a range for task performance or effect sizes of stimulation. Therefore, we estimated a sample size based on our two previous studies which showed significant effects of a-tDCS on motor task performance in samples of 40 subjects in a non-dominant hand tweezer pegboard task and of 58 subjects in a non-dominant hand, dart throwing task. Notably, between the dart and tweezer task, the dart task was characterized by more degrees of freedom and, in turn, more task performance variability. Since the Step Mania task involved less upper arm movement and fewer degrees of freedom than the dart task but more cognitive and technical complexity than the tweezer task, we determined that an appropriate recruitment target would likely lay somewhere between the 40 and 58 subjects. Still, because we were unsure, we opted to recruit 60 with the hopes that this number would provide us with sufficient statistical power for our analyses.

Could the authors provide clarification on whether the study was retrospective? Additionally, it would be valuable to know if the data analyzed in this study has also been considered in another publication.

This study was prospective, and the data has not been considered in another publication.

It would be appreciated if the authors could provide details on the randomization procedure and specify if participation was voluntary. Consider following guidelines for reporting your experiment.

Using MathWorks MATLAB software, a randomly generated number sequence was created and used during screening to assign subjects into either the anodal tDCS group (a-tDCS) or SHAM stimulation group (SHAM) based upon their order of recruitment.

Could the authors enhance the resolution of Figure 1?

Per your suggestion, we have resubmitted Figure 1 in higher resolution.

An explanation of the rationale behind choosing current intensity and stimulation duration would be helpful.

Added the following to the Methods section under the subheading, tDCS; 

“These parameters were chosen because previous research, including work from our lab, has shown that 1mA of current delivered for 20 minutes is sufficient to elicit changes in cortical excitability and motor performance and that higher doses (longer durations and/or higher intensities) may result in diminishing or inverse effects (Batsikadze et al., 2013; Esmaeilpour et al., 2018; A. W. Meek et al., 2021; Rawji et al., 2018; M. A. Wilson et al., 2022).”

Could the authors provide clarification on whether participants' naivety to rhythm games served as a control variable in the study? If not, I think it would be beneficial to include a comment on this aspect in the discussions. However, in the discussion (pag 23), it is mentioned that participants were naive to rhythm games – was this an exclusion criterion then? If so, it would be helpful to add this information to the description of exclusion criteria.

I believe our phrasing was a bit confusing on this topic. Subjects were naïve to playing rhythm games with their non-dominant hand, not necessarily naïve to rhythm games entirely. All subjects reported having never played a unimanual rhythm game (like Step Mania) with their non-dominant hand. This point has been reworded for better clarity and to note that subjects’ prior exposure to these types games may have influenced their ability to learn with the non-dominant hand.

Results:

Consider incorporating the results of the post-test into Figure 2 to provide an overarching view of the experiment and the longevity of effects post-training.

Figures 2 and 3 have been combined so that the post-test results are displayed within the context of the rest of the experiment.

Discussion:

The discussion section of the paper addresses various crucial points regarding the outcomes derived from the application of M1 a-tDCS in acquiring motor skills within a timing-based video game. Here are some suggestions to further elaborate on the discussion:

Elaborate on the possibility that the a-tDCS group started with more errors in baseline in Figure 4C.

This is a good point. We acknowledge that this difference in KER appears significant, however, it doesn’t seem to have impacted our results. In an attempt to provide more clarity, we’ve added the following sentence to the discussion section addressing this point; “Despite what appear to be disparities in baseline KER scores between groups (Fig. 4C), these differences were non-significant (p = 0.149), resulted in only marginal (∼7 points) differences in the baseline PI scores between groups, and were largely attenuated by the end of the initial practice block.”

Discuss additional limitations and suggest areas for future research.

Given that the study's sample comprises a healthy population, consider rephrasing the last sentence of the discussion (page 24, l. 14-16) to emphasize the need for additional research within a rehabilitation context.

We have revised the final paragraph of the discussion to include a better summary of the study’s limitations. Additionally, a sentence about the need for research within a rehab context has been added.

Minor comments:

Italicize the p-values

Spell out PM and SMA before using the acronyms (pag 22)

Both of these have been addressed.

---

## [Editor Report · Decision Letter 1]

16 May 2024

Anodal M1 tDCS enhances online learning of rhythmic timing videogame skill

PONE-D-23-37863R1

Dear Dr. Greenwell,

We’re pleased to inform you that your manuscript has been judged scientifically suitable for publication and will be formally accepted for publication once it meets all outstanding technical requirements.

Kind regards,

Sandra Carvalho, Ph.D.

Academic Editor

PLOS ONE

---

## [Editor Report · Acceptance letter]

24 May 2024

PONE-D-23-37863R1 

PLOS ONE

Dear Dr. Greenwell, 

I'm pleased to inform you that your manuscript has been deemed suitable for publication in PLOS ONE. Congratulations! Your manuscript is now being handed over to our production team.

Kind regards, 

on behalf of

Professor Sandra Carvalho 

Academic Editor

PLOS ONE